# Are gamers better laparoscopic surgeons? Impact of gaming skills on laparoscopic performance in "Generation Y" students

Rabi Datta[1◉], Seung-Hun Chon[1◉*], Thomas Dratsch[2], Ferdinand Timmermann[2], Luise Müller[2], Patrick Sven Plum[1], Stefan Haneder[3], Daniel Pinto dos Santos[3], Martin Richard Späth[4,5], Roger Wahba[1], Christiane Josephine Bruns[1], Robert Kleinert[1]

1 Department of General, Visceral, Cancer, and Transplant Surgery, University Hospital of Cologne, Cologne, Germany, 2 University of Cologne, Cologne, Germany, 3 Institute of Diagnostic and Interventional Radiology, University Hospital of Cologne, Cologne, Germany, 4 Department II of Internal Medicine and Center for Molecular Medicine, University of Cologne, Cologne, Germany, 5 Cologne Excellence Cluster on Cellular Stress Responses in Aging Associated Diseases (CECAD), University of Cologne, Cologne, Germany

◉ These authors contributed equally to this work.
* seung-hun.chon@uk-koeln.de

**Data Availability Statement:** Data are available on OSF using the following link: https://osf.io/zg2ku/?view_only=23e138c4c9f94bc9848181e0d680fb0b.

## Abstract

### Background

Both laparoscopic surgery and computer games make similar demands on eye-hand coordination and visuospatial cognitive ability. A possible connection between both areas could be used for the recruitment and training of future surgery residents.

### Aim

The goal of this study was to investigate whether gaming skills are associated with better laparoscopic performance in medical students.

### Methods

135 medical students (55 males, 80 females) participated in an experimental study. Students completed three laparoscopic tasks (rope pass, paper cut, and peg transfer) and played two custom-designed video games (2D and 3D game) that had been previously validated in a group of casual and professional gamers.

### Results

There was a small significant correlation between performance on the rope pass task and the 3D game, Kendall's $\tau(111) = -.151$, $P = .019$. There was also a small significant correlation between the paper cut task and points in the 2D game, Kendall's $\tau(102) = -.180$, $P = .008$. Overall laparoscopic performance was also significantly correlated with both the 3D game, Kendall's $\tau(112) = -.134$, $P = .036$, and points in the 2D game, Kendall's $\tau(113) = -.163$, $P = .011$. However, there was no significant correlation between the peg transfer task and both games (2D and 3D game), $P = $ n.s..

**Funding:** The author(s) received no specific funding for this work.

**Competing interests:** No authors have competing interests.

## Conclusion

This study provides further evidence that gaming skills may be an advantage when learning laparoscopic surgery.

## Introduction

Laparoscopic surgery has become the gold standard for many surgical procedures and new technical developments are pushing the boundaries of the possible, extending the scope of this field to more complex surgical procedures [1, 2]. However, compared to open surgery, laparoscopic surgery does take surgeons in training significantly longer to master, and can make already challenging operations even more difficult for novices [3, 4]. Known factors contributing to the difficulty of laparoscopic surgery are unfamiliar hand movements, the reduction from real-life three-dimensional (3D) stereoscopic vision to a virtual two-dimensional (2D) image, and the transfer from a familiar field of vision to the distorted picture of the laparoscopic camera. Another domain that also trains these particular skills is video games; however, the impact of video gaming skills on laparoscopic skills is still open to debate. Although it is has been shown that gaming experience has a positive effect on eye-hand coordination and also modulates different aspects of visuospatial ability, spatial resolution and cognitive flexibility, the real effect of gaming skills on laparoscopic performance is still unknown [5–10]. While some authors reject a correlation, others describe a positive association [11–13]. However, the majority of these studies are observational and hence information about real causation remains unclear [14]. Most studies equate "gaming experience" with "gaming skills", which is inaccurate as there is no validated, standardized definition of "gaming experience" [15]. The underlying questionnaires are not standardized, nor are the definitions of "game experience", ranging from self-evaluation (non-gamer / novice / expert) to amount of time spent gaming [11, 16–18]. Previous studies have small sample sizes and report heterogeneous results because there is no standardized system for scoring laparoscopy results [19, 20]. Furthermore, demographic changes have led to a new population of students (Generation Y) and future residents who have had more gaming experience when growing up and were surrounded by computers and smartphones [21]. It has been shown that this generation is more proficient when it comes to gaming than previous generations [22, 23].

There is currently no study available that assesses the gaming skills of the current generation of students with a validated test method and correlates these results with laparoscopic performance. This study can therefore address the question whether gaming skills are a predictor for a steeper learning curve in laparoscopic surgery.

The goal of this study was therefore twofold:

Firstly, to create a valid measure of gaming skills, we developed two custom-designed video games (one 2D and one 3D game) that were validated in a group of professional and casual gamers.

Secondly, to investigate whether gaming skills are associated with laparoscopic skills in medical students, medical students first performed three laparoscopic tasks and then played the two validated video games.

## Material and methods

### Ethics

Ethics Committee approval was obtained before the study (Ethics Committee, University of Cologne) and the current study adheres to the criteria of our local ethics committee (No. 18–176). Written informed consent was given by all subjects before study inclusion.

### Definition of the requirements for custom video games

To test the participants' gaming skills, two custom video games (one 2D game and one 3D game) were developed for this study. Features of the games should be based on established gaming concepts (i.e., "jump and run", "side scroller") that experienced players (EP) are familiar with whereas the games themselves should be totally new to all participants [24]. Unity3D was chosen as the development environment for the games. One requirement was that the games should be enjoyable and easy to learn in order to motivate students, which was achieved by using established techniques of modern game design [25]. Furthermore, the games should be "hard to master" in order to further distinguish between experienced players (EP) and non-experienced players (NEP). This was achieved by including a hidden goal ("collecting power-ups") in the game that was less apparent than the obvious goal ("surviving as long as possible"). In addition, we created a stressful gaming environment by constantly increasing the games' speed and by including fast and stressful music [26].

In order to measure different aspects of gaming skills, two games were developed: one 2D game and one 3D game. The 2D game tests eye-hand coordination and was designed as a classic "side-scroller" game, in which the player navigates a "spaceship" through a two-dimensional virtual environment that scrolls from the right- to the left-hand side of the screen at a fixed speed (see Fig 1). The player controls the spaceship with the mouse similar to moving the cursor on a computer. The main goal of the game is to avoid a collision with any object for as long as possible ("to survive"), which tests spatial resolution [9]. When the spaceship collides with other objects (i.e., rocks, other spaceships, or bullets), the game is over. The spaceship

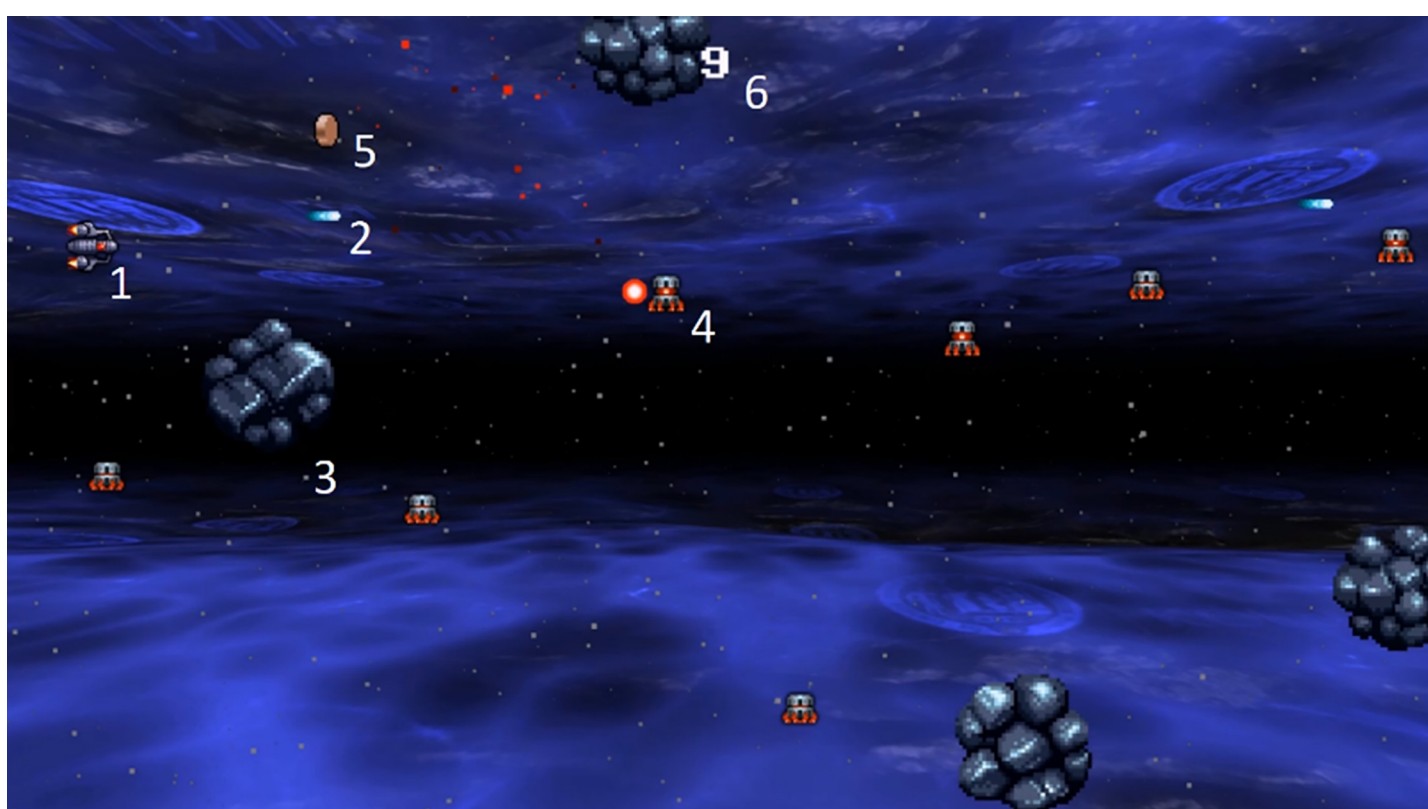

**Fig 1.  "Side-scroller" game where the player navigates a spaceship (1) that shoots (2).** The main goal is to avoid a collision with objects (3) and other spaceships (4), which also shoot bullets. Special objects (5) can be collected to gain more power. Destroyed objects earn points (6).

shoots automatically and bullets can destroy other objects in front of the player to gain points. In addition, several power-ups add improvements to the ship (i.e., faster shooting) and can be collected by flying over them. This "hidden goal" was not communicated before the experiment and measures "game understanding" as part of cognitive flexibility [10].

To test visuospatial ability, we created a 3D game with simple controls (only left and right arrows). Again, the player controls a "spaceship" but this time in three-dimensional space. The ship automatically flies either on the inside or outside of a tube, always sticking to the wall of the tube. The only task is to avoid collision with obstacles and to survive as long as possible. Participants can rotate the tube to the left or right in order to avoid collision. Difficulty increases from level to level as the speed of the spaceship increases and obstacles start to move (see Fig 2).

Player performance was measured using the following parameters: In both the 2D and the 3D game time "survived" was used as a general measure of gaming experience. Additionally, in the 2D game, analytical thinking was determined by counting the absolute number of collected power-ups because this feature of the game was considered only to be obvious to experienced players. It was hypothesized that experienced players would survive longer in both games (2D and 3D game) and collect more points in the 3D game.

The games were validated in a separate sample consisting of professional gamers (ProG: 29 males, 6 females; $Mean_{Age} = 22.23$, $SD_{Age} = 2.78$) and casual gamers (CasG: 27 males, 8 females; $Mean_{Age} = 25.14$, $SD_{Age} = 4.87$). Both groups played five rounds of each game. Players in the professional gamer group were members of one of Europe's "Electronic Sports Leagues" (ESL) and had participated in at least 10 E-Sports Tournaments. All these players were "full time" gamers who had trained with the game "Counter Strike" for at least 60 to 90 hours per week for several years. Counter strike is known as a useful training tool to increase fine motor skills and movement coordination [27]. The group of casual gamers was comprised of players that did not play professionally and did not exceed more than two hours of gaming peer week.

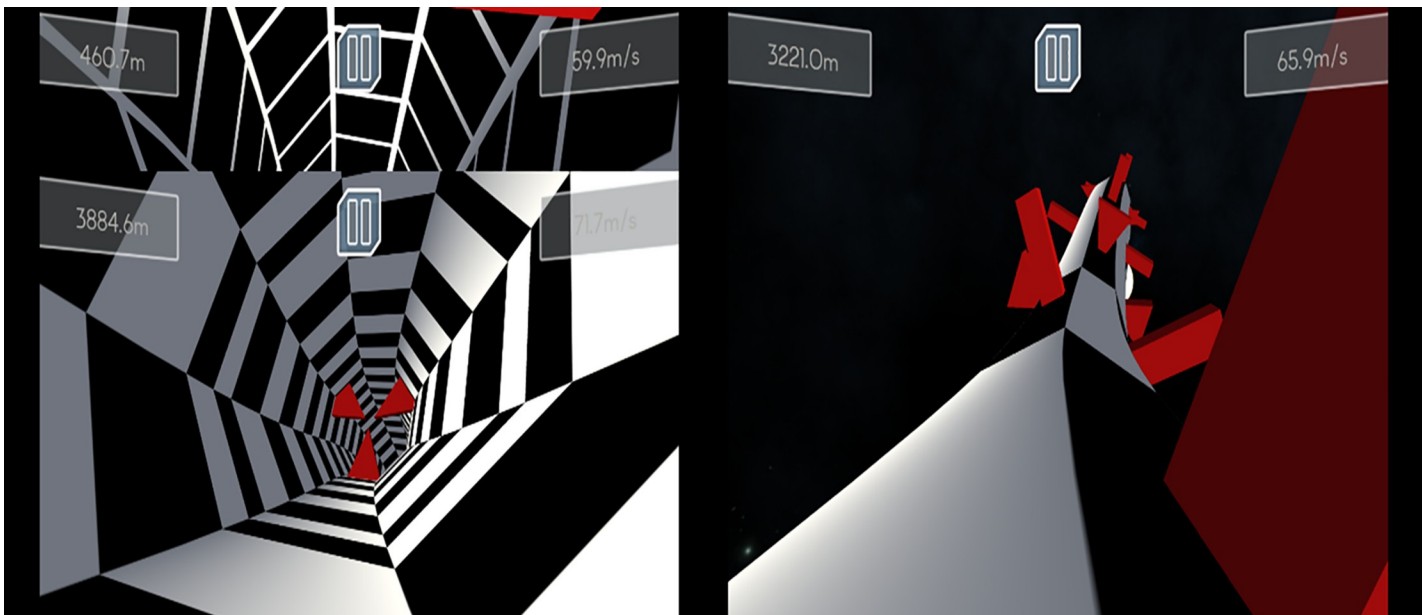

**Fig 2. Concept of the 3D game: The player controls a space ship from a first-person perspective flying through the inside of a tube or on the outside of a tube.**

## Validation of custom games as test tools

Construct validity for the games was tested by comparing professional and casual gamers. In the 2D game, professional gamers survived significantly longer (M = 38 seconds, SD = 12 seconds) than casual gamers (M = 21 seconds, SD = 10 seconds), $p < .005$. Professional gamers also collected significantly more power-ups (M = 9.0, SD = 1.2) than casual gamers (M = 4.0, SD = 0.8), $p < .005$, indicating a higher level of game understanding. In the 3D game, professional gamers also survived significantly longer (M = 7225, SD = 1268) than casual gamers (M = 4200, SD = 1668), $p < .005$, indicating better visuospatial ability.

As the results of the validation study show, players with high gaming skills (professional gamers) did perform significantly better in both games than people with low gaming skills (casual gamers). It is therefore safe to conclude that performance in the two games does correlate with actual gaming skills. Thus, the two games can be used, firstly, as a valid measuring tool to assess gaming skills in a sample of medical students and, secondly, to assess whether gaming skills are correlated with laparoscopic performance.

## Laparoscopy

Laparoscopic skills were measured using a laparoscopic training simulator (eoSim, eoSurgical Ltd, Edinburgh, UK). Laparoscopic procedures were recorded via an integrated video camera system that was connected to a standard tablet computer.

The following laparoscopic tasks were used to measure laparoscopic skills: rope pass, paper cut, and pegboard transfer. These laparoscopic tasks were selected because they have been used as valid measurement tools in several prior studies [28, 29]. The three tasks are depicted in Fig 3.

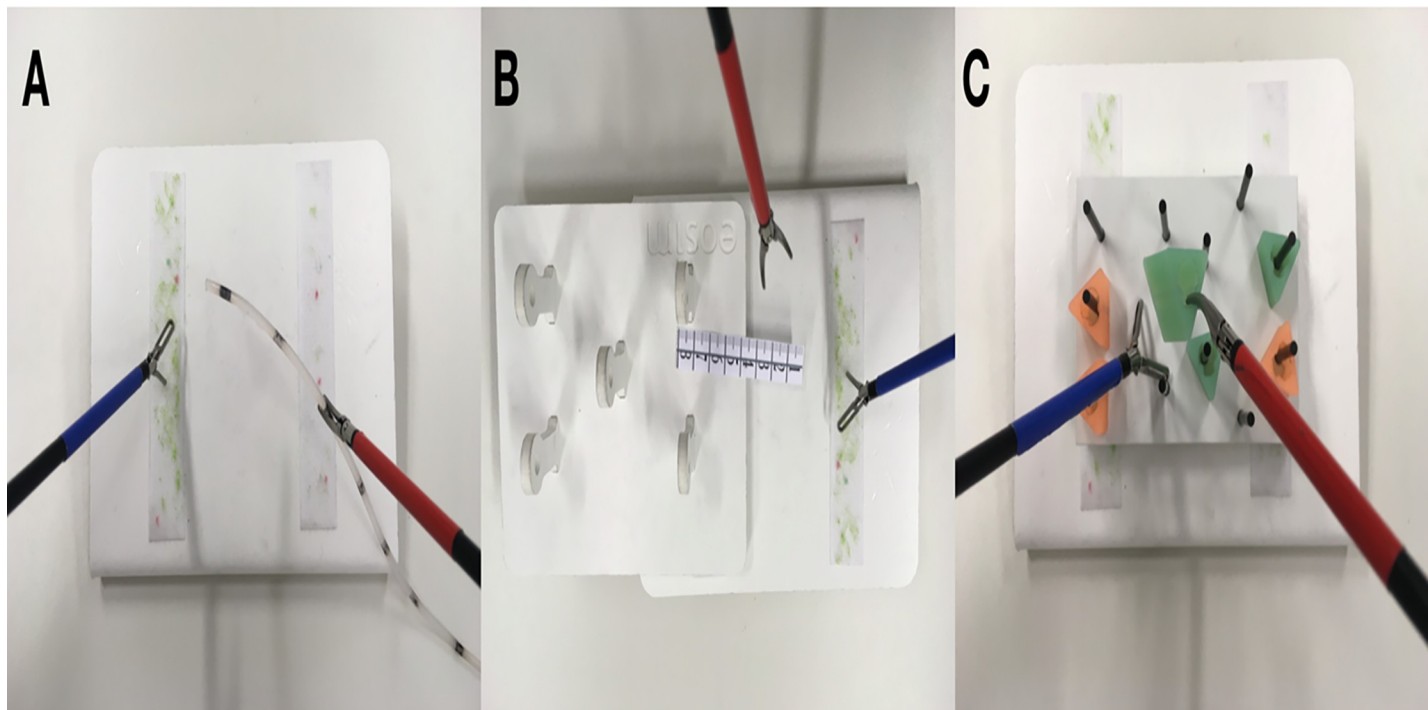

**Fig 3.** Rope pass (A), paper cut (B), pegboard transfer (C).

In the rope pass task, students' task was to pass a 30 cm long silicone tube from one instrument to the other, while only touching the tube at certain marked areas (size of each area was 3mm; the space between each area was 3 cm). Touching the silicone tube at a non-marked area was counted as an error.

In the paper cut task, students were presented with an 8 cm long paper ruler with markings every millimeter and every centimeter. Students' task was to cut along the markings without fully cutting the paper ruler in half. Cutting through the paper or cutting in a non-marked area was counted as an error.

In the pegboard transfer task, students were presented with a pegboard with 11 metal rods and six triangles. Students' task was to transfer the triangles between the metal rods. Incorrect placement of the triangles was counted as an error.

Students watched an instructional video explaining the three laparoscopic tasks. After that, students received a handout describing all three tasks in more detail. Students had to complete each laparoscopic task three times. All laparoscopic tasks were recorded on video. For each task, time to complete and number of errors was measured based on the videos. The beginning of each trial was defined as the moment when the students first touched the materials of the task at hand (rope, paper, or triangle) with the laparoscopic instruments. The end of each trial was defined as the moment when students had completed the task and had released the laparoscopic instruments onto the floor of the laparoscopic training box.

## Questionnaires

After participants had completed the laparoscopic tasks and the two video games, they completed the NASA task load index (NASA-TLX; https://humansystems.arc.nasa.gov/groups/TLX/). Additionally, students were asked whether they own a gaming console, for how many years they had been playing video games, and how many hours of video games they used to play per day.

## Procedure

Two students participated in each testing session. At the beginning of the experiment, students were greeted by the experimenter and sat down in front of a computer. Students then watched an instructional video describing the three laparoscopic tasks. After that, they received a handout describing the tasks. Then each student was assigned their own laparoscopic box and started with the first trial of the first task. Students always completed the three laparoscopic tasks in the following order: rope pass, paper cut, and pegboard transfer. Each laparoscopic task was performed three times by each student. All laparoscopic tasks were recorded on video. After students had completed the laparoscopic tasks, they sat down in front of a computer to play the two computer games. Students first played five rounds of the 2D game and then played one round of the 3D game. All gaming sessions were also recorded on video for further analysis. After students had finished playing the video games, they completed several questions about their prior gaming experience and the NASA task load index (NASA-TLX).

## Group distribution

One hundred and thirty-five medical students (55 males, 80 females; mean age = 23.66, age range: 20–33) were recruited at the University Hospital of Cologne through mailing lists, flyers, and social networks. The study was conducted between December 2018 and February 2019. The inclusion criteria were the following: Students had to be enrolled as medical students at the University of Cologne. For detailed demographic information see Table 1.

Table 1. Demographic data.

|  | Male | Female |
|---|---|---|
| **n** | 55 | 80 |
| **Age (Mean, SD)** | 24.42 (2.78) | 23.14 (3.17) |
| **Handedness (left:right)** | 4:51 | 7:73 |
| **Glasses (yes:no)** | 21:34 | 47:33 |
| **Semester** |  |  |
| 1. Preclinical | 2 | 11 |
| 2. Preclinical | 2 | 2 |
| 3. Preclinical | 3 | 5 |
| 4. Preclinical | 2 | 4 |
| 1. Clinical | 4 | 9 |
| 2. Clinical | 12 | 16 |
| 3. Clinical | 1 | 6 |
| 4. Clinical | 10 | 14 |
| 5. Clinical | 2 | 3 |
| PJ | 17 | 10 |

## Statistical analysis

For each of the three laparoscopic tasks time to complete and number of errors were recorded. In line with Rosser et al., for each error 5 seconds were added to the time to complete the task [30]. This combined measurement was used in all statistical analyses. A statistical power analysis was performed for sample size estimation. Because the implementation of new tools into the medical curriculum can be expensive and time consuming, potential new methods should have a sufficiently large benefit. However, promising new tools should also not be overlooked. Therefore, as a compromise, our study should be sufficiently powered to detect medium sized-effects for the within-group comparisons. With an alpha = .05 and power = .80, the projected sample size needed to detect a medium effect for the within-group comparisons was N = 90 (GPower 3.1). With regard to correlations, a minimum sample size of N = 84 was needed to detect a medium-sized effect (GPower 3.1). Data were analyzed using the Statistical Package for the Social Sciences (SPSS, Version 25; IBM, 2017). Group comparisons were conducted using t tests and mixed ANOVAs. Kendall's τ was used as a robust measure of correlation.

## Results

### Laparoscopic performance

**Learning effect.** To test whether laparoscopic performance improved over the three trials and whether the laparoscopic tasks differed in difficulty, we conducted a 3 × 3 mixed ANOVA (Trial × Task). There was a significant main effect for trial, $F(2, 56) = 56.74$, $p < .001$, $\eta^2_p =$ .670, indicating that students' laparoscopic performance improved from the first to the third trial. There was also a significant main effect for task, $F(2, 56) = 24.90$, $p < .001$, $\eta^2_p = .471$, indicating that the tasks differed in difficulty. There was also a significant interaction effect, $F(4, 112) = 8.33.90$, $p < .001$, $\eta^2_p = .229$, indicating that the differences in difficulty of the tasks varied over the three trials (see Fig 4).

**Gender differences in laparoscopic performance.** To test whether male and female participants differed in laparoscopic performance, we conducted a 2 × 3 mixed ANOVA (Gender × Task). There was a significant main effect for task, $F(2, 208) = 74.10$, $p < .001$, $\eta^2_p =$ .416, mirroring the results from the first analysis that the three laparoscopic tasks differed in

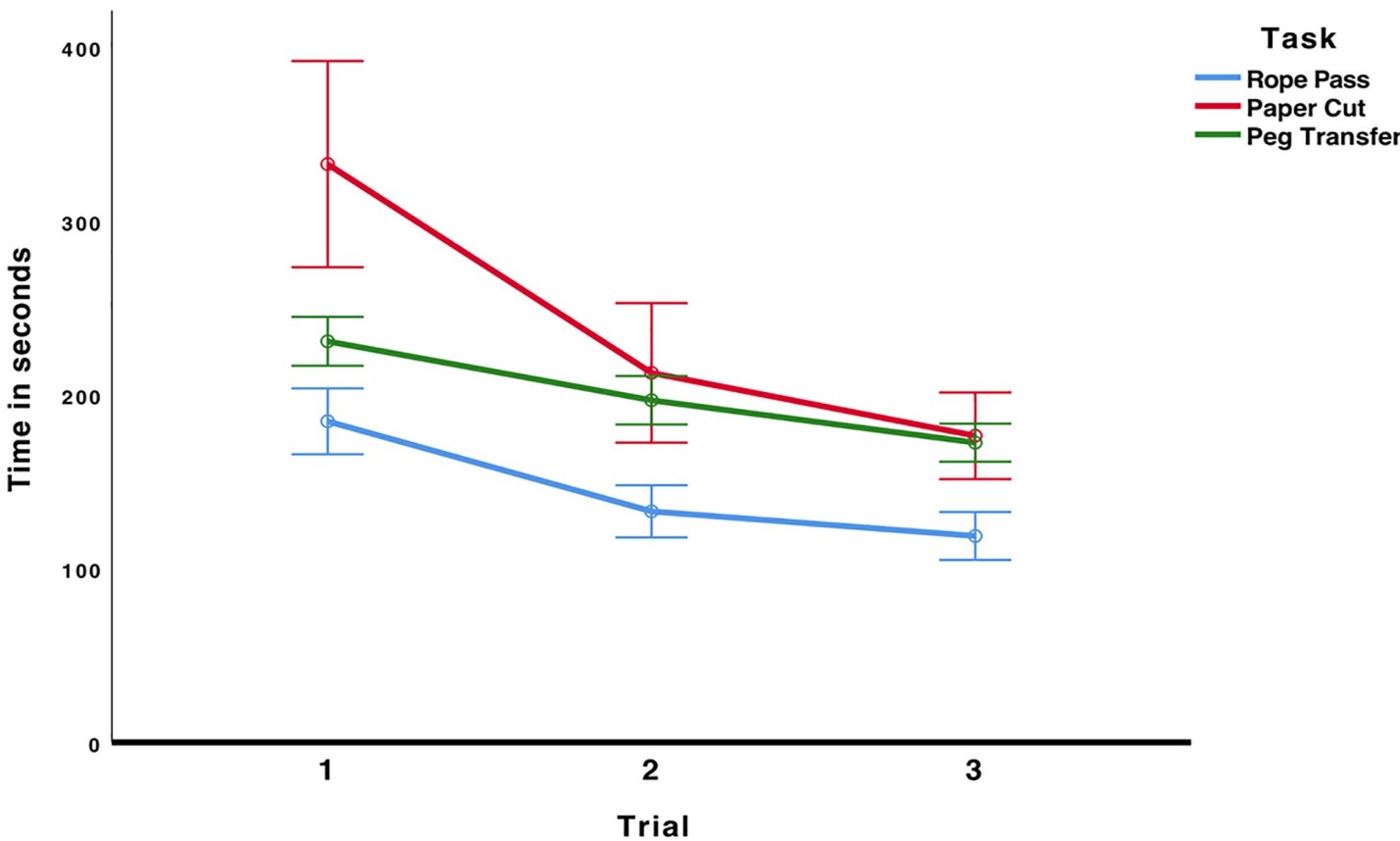

**Fig 4. Time to complete each laparoscopic task.** Error bars represent 95%-CI of the mean.

difficulty. However, there were no significant main or interaction effects for gender, all $F < 1.43$, indicating that male and female participants did not differ in laparoscopic performance.

**Correlations between laparoscopic tasks.** All laparoscopic tasks were significantly correlated. There was a significant correlation between the tasks rope pass and paper cut, Kendall's $\tau(118) = .172$, $P = .006$. There was also a significant correlation between the tasks rope pass and peg transfer, Kendall's $\tau(116) = .177$, $P = .005$. The tasks paper cut and peg transfer were also significantly correlated, Kendall's $\tau(116) = .141$, $P = .033$. This suggests that students who performed better on one laparoscopic task also performed better on the other laparoscopic tasks.

**Correlations between measurements of gaming skills.** To test whether there were significant relationships between the different measurements of gaming skills and the results from the two games, we correlated the questions about prior gaming experience and the results from the two games (2D and 3D Game). There was a significant correlation between the results of the 3D game and past gaming experience (hours of gaming per day), Kendall's $\tau(117) = .289$, $P < .001$. Time in the 2D game, Kendall's $\tau(118) = .229$, $P = .001$, and points in the 2D game, Kendall's $\tau(118) = .320$, $P < .001$, were also both significantly correlated with past gaming experience (hours of gaming per day). Additionally, ownership of a gaming console was also significantly correlated with results of the 3D game, Kendall's $\tau(117) = .158$, $P = .038$, time in the 2D game, Kendall's $\tau(118) = .173$, $P = .023$, and points in the 2D game, Kendall's $\tau(118) = .182$, $P = .017$. However, there were no significant correlations between years of gaming

experience and any of the gaming results. Overall, this suggests that there was a positive correlation between self-reported prior gaming experience and gaming skills in the two games.

**Correlations between laparoscopic performance and gaming.** To test whether there were significant relationships between laparoscopic performance and gaming, we correlated the average time to complete each laparoscopic task and the results from the two games (2D and 3D game). For results see Table 2. Students who achieved more points in the 3D game also performed the Rope Pass Task more quickly. In addition, students who achieved more points in the 2D Game also performed the paper cut task more quickly. Lastly, overall laparoscopic performance was significantly correlated with performance in the 3D game and points in the 2D game.

**Learning curve.** To test whether the overall relationship between average time to complete each laparoscopic task and the results of the games held true for each of the individual three trials per task, we also correlated the relationship between each individual trial and the results of the two games (2D and 3D game). For results see Table 2. Students who achieved more points in the 3D game also performed the Rope Pass Task more quickly in the 1st and 3rd trial. In addition, students who achieved more points in the 2D game also performed the paper cut task more quickly in the 1st trial.

**Correlations between laparoscopic performance and measurements of gaming experience.** More years of gaming experience were associated with a significantly better performance in the paper cut task, Kendall's $\tau(107) = -.146$, $P = .031$. Ownership of a gaming console was also associated with better overall laparoscopic performance, Kendall's $\tau(128) = -.160$, $P = .028$.

**Gender differences in gaming.** To test whether male and female students differed in gaming performance, we conducted three different $t$ tests. Male students (M = 4851, SD = 2377) achieved significantly more points in the 3D game than female students (M = 3729, SD = 2303), $t(115) = 2.54$, $p = .012$, $d = 0.48$. Male students (M = 23.9, SD = 11.1) also lasted significantly longer in the 2D game than female students (M = 19.3, SD = 7.9), $t(76) = 2.45$, $p = .017$, $d = 0.48$. Male students (M = 11.1, SD = 12.0) also achieved significantly more points in the 2D game than female students (M = 6.4, SD = 5.4), $t(58) = 2.48$, $p = .016$, $d = 0.54$.

**Effects of video games on laparoscopic performance.** In this study, participants first completed the laparoscopic tasks and then played the two video games. However, 21

**Table 2. Correlations between (Kendall's τ) laparoscopic performance and gaming $p < .05$ * $p < .01$.**

|  | 3D | 2D | 2D |
|---|---|---|---|
|  | Game | Game Time | Game Points |
| **1. Trial Rope Pass** | **-.131**$^*$ | -.054 | -.024 |
| **2. Trial Rope Pass** | -.071 | .033 | -.001 |
| **3. Trial Rope Pass** | **-.147**$^*$ | .025 | -.060 |
| **Rope Pass Overall** | **-.151**$^*$ | -.005 | -.064 |
| **1. Trial Paper Cut** | -.035 | -.101 | **-.185**$^*$ |
| **2. Trial Paper Cut** | -.100 | .041 | -.086 |
| **3. Trial Paper Cut** | -.091 | -.065 | -.098 |
| **Paper Cut Overall** | -.074 | -.095 | **-.180**$^{**}$ |
| **1. Trial Peg Transfer** | -.025 | .063 | -.023 |
| **2. Trial Peg Transfer** | -.080 | .035 | -.083 |
| **3. Trial Peg Transfer** | -.059 | .056 | -.039 |
| **Peg Transfer Overall** | -.087 | .039 | -.086 |
| **Overall Laparoscopic Performance** | **-.134**$^*$ | -.062 | **-.163**$^*$ |

participants first played the two video games and then performed the laparoscopic tasks. To test whether this reversed order had any effect on laparoscopic performance, we performed three separate ANOVAs comparing participants who had first completed the laparoscopic tasks with participants who had first played the video games. There was no significant difference in performance on the laparoscopic tasks between both groups, all $F < 1.06$.

**Other effects on performance.** Wearing glasses had no significant effect on gaming or laparoscopic performance, all $t < 1.29$. Students were categorized into three different groups (preclinical, clinical, and practical year) according to their level of clinical training. There was no significant effect of level of clinical training on gaming performance, all $F < 1.88$. Students of different levels of clinical training did not differ significantly on the rope pass and on the paper cut task, all $F < 1.93$. There was, however, a significant main effect for clinical training on the peg transfer task, $F(2, 114) = 3.31$, $p = .04$, $\eta^2_p = .055$. Pairwise comparisons revealed that students in their practical year (M = 208, SD = 40) completed the task significantly faster than students in the clinical part of their studies (M = 228, SD = 35), $t(89) = 2.31$, $p = .023$, $d = 0.53$. There was, however, no significant difference between students in the practical year and students in the preclinical part of their studies, $t < .587$.

**Correlations between NASA task load scale and laparoscopic performance.** Students who rated their performance to be better on the NASA task load scale also performed significantly faster in the rope pass task, Kendall's $\tau(127) = -.222$, $P = .001$, and the peg transfer task, Kendall's $\tau(117) = -.168$, $P = .012$.

## Discussion

The goal of this study was to investigate whether gaming skills are associated with laparoscopic performance in medical students. As the results show, students who performed the rope pass task more quickly also achieved more points in the 3D game. In addition, students who performed the paper cut task more quickly also achieved more points in the 2D game. Lastly, overall laparoscopic performance was significantly correlated with performance in the 3D game and points in the 2D game.

It is interesting to note that most indirect measures of gaming skills (questionnaire data) were not correlated with laparoscopic performance—even though direct (2D and 3D game) and indirect (questionnaire data) measures were also correlated. This underscores the importance of using actual video games to measure gaming skills instead of just using questionnaires. General questions about gaming habits may be too broad and are possibly not able to capture the practical aspects of gaming experience that are relevant to laparoscopic performance. However, two indirect measures of gaming skills (ownership of a gaming console and years of gaming) were also correlated with laparoscopic performance. Whether these correlations reflect actual associations or just random correlations has to be established in future studies.

Not considering the main focus of the study, it is encouraging to see that all participants—regardless of prior gaming skills—were able to improve their performance over the three trials per task, underlining the fact that laparoscopic performance is a skill that improves with practice. This result is in line with prior research that shows that undergraduate students can significantly improve their laparoscopic surgery skills over a training period of 10 hours [31]. With regard to the Rope Pass Task, we also found a significant correlation between the results of the 3D game and laparoscopic performance in the 3rd trial, indicating that even when performing the task for the third time students who achieved more points in the 3D game performed the task more quickly than students who achieved less points in the 3D game. Whether this advantage of gaming skills would still be relevant after a longer training period needs to be investigated in future studies.

Additionally, all three laparoscopic tasks were significantly correlated, suggesting that students who performed better on one laparoscopic task also performed better on the other laparoscopic tasks.

We also did find a positive correlation between the self-assessment part of the NASA task load scale and laparoscopic performance, indicating that students were able to assess their own performance on the laparoscopic tasks. This has important implications for the training of students because a feeling of competence has been shown to have a positive effect on learning.

Interestingly, we did also find a significant difference between male and female students in gaming performance. Male students performed better on both the 2D and the 3D game than female students. However, even though some male students seem to have more gaming skills than female students, this advantage did not translate into better overall laparoscopic performance of male students. This may be due to the fact that successful laparoscopic performance may rely on several other skills independent of gaming skills so that gaming skills do not guarantee successful laparoscopic performance. As a whole, our study was not designed to investigate gender differences in the relationship between gaming skills and laparoscopic performance and thus did not include an equal number of male and female students (55 males, 80 females). This is one limitation and future studies should further investigate specific gender effects in a larger sample of both male and female students.

Another factor influencing laparoscopic performance may be the amount of clinical training the medical students had received. Overall, we did not find that students in the later stages of their studies outperformed younger students on the laparoscopic tasks. However, considering the surgical curriculum at German universities, this result is not surprising because laparoscopic training is generally not a part of the medical curriculum.

It is important to note that the relationship between gaming and laparoscopic performance in our study was small. Based on our results, recruiting future surgical residents based solely on their gaming skills may not be warranted. However, as the small correlation shows, gaming skills should not be considered to be a negative factor. Additionally, using video games to train laparoscopic skills may not be the most efficient method because the transfer of skills from one task to the other may be small. As the increase in performance from the first to the third trial on all laparoscopic tasks in our study shows, the largest gains in laparoscopic skills are expected to be made through extended practice with the laparoscopic simulator. Thus, the most efficient way to prepare young surgery residents for the operating room is through practice with laparoscopic simulators.

To our knowledge, this is the first study that tests gaming skills with custom-designed games. This procedure checks for the confounding factor that experienced players may be familiar with the games used for testing [32]. The main goal of the two custom games used in the present study was to create two valid tools to measure gaming skill. The two games had to fulfill the following criteria: The games should be "hard to master" in order to distinguish between experienced players (EP) and non-experienced players (NEP). The games should increase in difficulty. This was achieved by creating a stressful gaming environment by constantly increasing the games' speed and by including fast music. The main purpose of the games was to measure gaming skill. As the validation with the professional gamers shows, we succeeded in creating two games that distinguish between experienced and non-experienced players. Thus, our games showed a positive validity for distinguishing between pro and casual gamers and hence were a suitable tool for measuring gaming skills.

Although the impact of playing video games on simulator performance has been reported by several authors, the causal factors that lead to an increase in performance are still unclear [33, 34]. Current experimental studies focus on training non-experience players (NEPs) over a period of time, while monitoring changes in laparoscopic performance and different cognitive

domains [14, 35]. However, it is known that repetition of a particular activity has a positive effect on performance of that activity and moreover a transfer of psychomotor skills from one activity to another activity is possible (Mozart Effect) [36]. More than that, the practical application of designing a game is questionable. Nowadays, virtual reality laparoscopic simulators are broadly available and also show a positive training effect on laparoscopic performance [37, 38]. Hence, unless training with specific games shows a superior effect on skills, the effort of designing and validating a specific game may be questionable.

Furthermore, future studies could also investigate the laparoscopic performance of professional gamers, such as E-Sports players. However, this was beyond the scope of this investigation. Because the E-Sports players were recruited at a special event, it was not possible for us to extend the present study with a group of E-sports players performing the laparoscopic tasks.

Finally, this study uses a simulator to assess laparoscopic skills; therefore, the reported relationship between laparoscopic performance and video game playing may only apply to simulated performance.

This study provides further evidence that gaming skills may be an advantage when learning laparoscopic surgery.

## Acknowledgments

We thank Ms. Claire Cahm for proofreading. We thank Ms. Amy Rambow and Mr. Juan Soriano Nunez for taking care of the students.

## Author Contributions

**Conceptualization:** Rabi Datta, Seung-Hun Chon, Luise Müller, Patrick Sven Plum, Christiane Josephine Bruns, Robert Kleinert.

**Data curation:** Rabi Datta, Seung-Hun Chon, Thomas Dratsch, Ferdinand Timmermann, Luise Müller, Patrick Sven Plum, Stefan Haneder, Daniel Pinto dos Santos, Martin Richard Späth, Roger Wahba, Christiane Josephine Bruns, Robert Kleinert.

**Formal analysis:** Rabi Datta, Seung-Hun Chon, Thomas Dratsch, Ferdinand Timmermann, Luise Müller, Stefan Haneder, Daniel Pinto dos Santos, Martin Richard Späth, Christiane Josephine Bruns, Robert Kleinert.

**Investigation:** Rabi Datta, Seung-Hun Chon, Thomas Dratsch, Robert Kleinert.

**Methodology:** Rabi Datta, Seung-Hun Chon, Thomas Dratsch, Daniel Pinto dos Santos, Roger Wahba.

**Project administration:** Rabi Datta, Seung-Hun Chon, Ferdinand Timmermann, Luise Müller, Christiane Josephine Bruns.

**Resources:** Rabi Datta, Seung-Hun Chon, Stefan Haneder, Martin Richard Späth.

**Software:** Rabi Datta, Seung-Hun Chon, Thomas Dratsch, Ferdinand Timmermann, Patrick Sven Plum, Stefan Haneder, Daniel Pinto dos Santos, Robert Kleinert.

**Supervision:** Rabi Datta, Seung-Hun Chon, Luise Müller, Martin Richard Späth, Christiane Josephine Bruns.

**Validation:** Rabi Datta, Seung-Hun Chon, Thomas Dratsch, Patrick Sven Plum, Daniel Pinto dos Santos, Robert Kleinert.

**Visualization:** Rabi Datta, Seung-Hun Chon, Ferdinand Timmermann, Roger Wahba.

**Writing – original draft:** Rabi Datta, Seung-Hun Chon, Thomas Dratsch, Ferdinand Timmermann, Luise Müller, Patrick Sven Plum, Stefan Haneder, Daniel Pinto dos Santos, Martin Richard Späth, Roger Wahba, Christiane Josephine Bruns, Robert Kleinert.

**Writing – review & editing:** Rabi Datta, Seung-Hun Chon, Thomas Dratsch, Ferdinand Timmermann, Luise Müller, Patrick Sven Plum, Stefan Haneder, Daniel Pinto dos Santos, Martin Richard Späth, Roger Wahba, Christiane Josephine Bruns, Robert Kleinert.

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
