## [Decision Letter · Decision Letter 0]

2 Oct 2019

PONE-D-19-23296

Are Gamers better laparoscopic surgeons? Impact of Gaming Skills on Laparoscopic Performance in “Generation Y” students

PLOS ONE

Dear Dr. Chon,

Thank you for submitting your manuscript to PLOS ONE. After careful consideration, we feel that it has merit but does not fully meet PLOS ONE’s publication criteria as it currently stands. Therefore, we invite you to submit a revised version of the manuscript that addresses the points raised during the review process.

We would appreciate receiving your revised manuscript by Nov 16 2019 11:59PM. To enhance the reproducibility of your results, we recommend that if applicable you deposit your laboratory protocols in protocols.io, where a protocol can be assigned its own identifier (DOI) such that it can be cited independently in the future. For instructions see: http://journals.plos.org/plosone/s/submission-guidelines#loc-laboratory-protocols

We look forward to receiving your revised manuscript.

Kind regards,

Chee Kong Chui, PhD

Academic Editor

PLOS ONE

Journal Requirements:

2.  Please include additional information regarding the questionnaires used in the study: a) for the questionnaire developed as part of this study, if it is not under a copyright more restrictive than CC-BY, please include a copy as Supporting Information; b) please provide a reference for the NASA-TLX questionnaire or a link from where it can be downloaded.

- In your Methods section, please provide additional information about the participant recruitment method, such as: a) the recruitment date range (month and year), b) a description of any inclusion/exclusion criteria that were applied to participant recruitment, and c) a description of how participants were recruited.

- PLOS ONE requires that authors of manuscripts in which software is a central part of the manuscript make all relevant software available without restrictions upon publication of the work. Authors must ensure that software remains usable over time regardless of versions or upgrades. If the original software is not able to be shared, authors must provide a reasonable facsimile. In this case, we think that the 2D and 3D games that were developed as part of this work should be shared accordingly. Please see our policies on sharing software for more information: https://journals.plos.org/plosone/s/materials-and-software-sharing#loc-sharing-software.

Additional Editor Comments (if provided):

This paper is about a study investigating a possible connection between laparoscopic surgery capabilities and the skill of computer games. The study used two custom made games (one 2D and one 3D spaceship centred game) and three laparoscopic tasks (rope pass, paper cut, and peg transfer) to measure an individual’s gaming and laparoscopic surgery capabilities respectively. The result is a finding of a correlation between the performance of the laparoscopic task and the games. The correlation which is not a strong one, concurs with earlier similar studies.

Major revision is required to address the questions raised by the reviewers. The authors need to describe how their studies are different from existing studies.

Reviewers' comments:

Reviewer's Responses to Questions

**Comments to the Author**

1. Is the manuscript technically sound, and do the data support the conclusions?

Reviewer #1: Yes

Reviewer #2: Partly

2. Has the statistical analysis been performed appropriately and rigorously? 

Reviewer #1: Yes

Reviewer #2: Yes

3. Have the authors made all data underlying the findings in their manuscript fully available?

Reviewer #1: Yes

Reviewer #2: No

4. Is the manuscript presented in an intelligible fashion and written in standard English?

Reviewer #1: Yes

Reviewer #2: Yes

5. Review Comments to the Author

Reviewer #1: My comments are in the attachment

Reviewer #2: This work proposes a method for measuring video gaming skill and investigates its correlation with laparoscopic skill. Two custom designed video games were proposed for evaluation of gaming skill, with their construct validity ascertained with professional and causal gamers. Subsequently, medical students were evaluated for their gaming skills with the two video games and a laparoscopic simulator was used to evaluate their laparoscopic performance. The results showed correlations that concurs with a number of previous studies in the literature.

Overall the manuscript is well written. However the contributions of the work is limited. The following comments are intended to assist in improving the work.

Throughout the manuscript, references are made to the terms “skill”, “experience” and “performance”. If they refer to the same characteristic, perhaps the authors would like to stick to a single term. Otherwise, defining each term would help with the clarity of the text. In a similar vein, the authors reference the inaccuracy of equating “gaming experience” with “gaming skill” in the introduction. However, the title of the manuscript references “gaming skill” while the text in the abstract makes references to “gaming experience”. Likewise in the discussion, these terms seem to be used interchangeably.

The choices in the type of games seem quite arbitrary. For example, why was a 2D side-scroller used and is this the best type of game to measure eye-hand coordination?

A number of known factors contributing to difficulty of laparoscopic surgery are inverted tool motion due to fulcrum effects of the tool ports, loss of 3D vision and disconnected viewpoints from the laparoscopic camera. Why wasn’t established games with similar difficulties utilized instead? Alternatively, using input devices emulating the difficulties present in laparoscopic surgery would have been more appropriate. For example, perhaps using a controller with dual thumbsticks instead of mouse/arrow keys could be more representative of their ability to adapt to hand-eye coordination skills.

More details of the background of the professional gamers should be included and controlled for. ESL spans a large range of game genres. A professional real-time strategy or a first person shooter gamer would have different skillsets that they each excels in and measure differently when tested with the authors’ games.

Were the students controlled for prior experience of laparoscopic surgery? For example, knowledge of laparoscopic surgery, its difficulties and possible experience through videos of procedures or even personal attempts with box trainers.

Please cite references in order of appearance in text - [39] in line 382 is referenced out of sequence. Reference [28] in line 429 may be inappropriate. Please ensure if the results of the referenced work supports the statement adequately. In-text citation of references [29-32] are missing.

6. PLOS authors have the option to publish the peer review history of their article (what does this mean?). If published, this will include your full peer review and any attached files.

Reviewer #1: Yes: Yin Jun Hao

Reviewer #2: No

---

## [Author Response · Author response to Decision Letter 0]

23 Dec 2019

Dear Prof. Kong Chui,

Thank you very much for giving us the opportunity to resubmit our revised version of the manuscript. Please also find attached a list of all the authors and their contributions to the manuscript. 

Response to Journal Requirements 

Requirement 1:

AUTHORS’ Response to Requirement 1:

The title page has been changed to meet the style requirements.

- - - - - - - - - - - - - - - - - - - - - - - - - - - - - - - - - - - - - - - - - - - - - - - - - - - - - - - - - - -

Requirement 2 Part 1:

Please include additional information regarding the questionnaires used in the study: a) for the questionnaire developed as part of this study, if it is not under a copyright more restrictive than CC-BY, please include a copy as Supporting Information; b) please provide a reference for the NASA-TLX questionnaire or a link from where it can be downloaded.

AUTHORS’ Response to Requirement 2 Part 1:

a) Students were asked whether they own a gaming console, for how many years they had been playing video games, and how many hours of video games they used to play per day. We added that information to the paper.

b) We added a link to the NASA-TLX questionnaire to the paper.

- - - - - - - - - - - - - - - - - - - - - - - - - - - - - - - - - - - - - - - - - - - - - - - - - - - - - - - - - - -

Requirement 2 Part 2:

In your Methods section, please provide additional information about the participant recruitment method, such as: a) the recruitment date range (month and year), b) a description of any inclusion/exclusion criteria that were applied to participant recruitment, and c) a description of how participants were recruited.

AUTHORS’ Response to Requirement 2 Part 1:

We added the following information to the paper: 

One hundred and thirty-five medical students (55 males, 80 females; mean age = 23.66, age range: 20–33) were recruited at the University Hospital of Cologne through mailing lists, flyers, and social networks. The study was conducted between … and … The inclusion criteria were the following: Students had to be enrolled as medical students at the University of Cologne.

- - - - - - - - - - - - - - - - - - - - - - - - - - - - - - - - - - - - - - - - - - - - - - - - - - - - - - - - - - -

Requirement 2 Part 3:

- PLOS ONE requires that authors of manuscripts in which software is a central part of the manuscript make all relevant software available without restrictions upon publication of the work. Authors must ensure that software remains usable over time regardless of versions or upgrades. If the original software is not able to be shared, authors must provide a reasonable facsimile. In this case, we think that the 2D and 3D games that were developed as part of this work should be shared accordingly. Please see our policies on sharing software for more information: https://journals.plos.org/plosone/s/materials-and-software-sharing#loc-sharing-software.

AUTHORS’ Response to Requirement 2 Part 3:

We will share the software.

- - - - - - - - - - - - - - - - - - - - - - - - - - - - - - - - - - - - - - - - - - - - - - - - - - - - - - - - - - -

Requirement 3:

AUTHORS’ Response to Requirement 3:

We will share a minimal anonymized data set necessary to replicate our study results.

- - - - - - - - - - - - - - - - - - - - - - - - - - - - - - - - - - - - - - - - - - - - - - - - - - - - - - - - - - -

Reviewer 1, Comment 1:

Factors surrounding the 2 methods of comparisons 

The types of games played in the Esports industry in the recent era consists heavily of Multiplayer Online Battle Arena games (e.g. Dota 2), Battle Royale games (e.g. 

PUBG) and First Person Shooting games (e.g. Overwatch). These games require one to have good action per minute capabilities and critical thinking within a short period of time. And this coincides with the needed skills to perform well for the custom games and players are given stressful conditions and the need to react to the obstacles which increases with speed throughout the games. Which are shown through the professional Esports players performing better in the custom games as compared to non-professionals. However, the skills required for laparoscopic tasks are mainly precision and visuospatial cognition. Thus, the custom games created may not be the best for this study. One suggestion could be changing the conditions of the custom game. For example, instead of making the game progressively harder through speed, the obstacles could instead get larger, making the area of allowable mistake smaller when passing though the obstacles. 

AUTHORS’ Response to Reviewer 1, Comment 1:

Reviewer 1 suggests that we should have used games that more closely resemble laparoscopic surgery. We agree that it would be interesting for future studies to investigate whether different types of games differ in their degree of correlation with laparoscopic performance. However, the main focus of our present study was to investigate whether gaming skill in general was positively correlated with laparoscopic performance. Therefore, the main goal of the two custom games used in the present study was to create two valid tools to measure gaming skill. The two games had two fulfill the following criteria:

- The games should be “hard to master” in order to distinguish between experienced players (EP) and non-experienced players (NEP).

- Increasing difficulty: This was achieved by creating a stressful gaming environment by constantly increasing the games’ speed and by including fast music.

The main purpose of the games was to measure gaming skill. As the validation with the professional gamers shows, we succeeded in creating two games that distinguish between experience and non-experience players.

- - - - - - - - - - - - - - - - - - - - - - - - - - - - - - - - - - - - - - - - - - - - - - - - - - - - - - - - - - -

Reviewer 1, Comment 2:

Purpose and aim of the study 

The purpose of the study was not coherent throughout the paper. It was discussed that the possibility of the connection between laparoscopic surgery and computer games could bring about a change in recruitment and training (background, line 28- 29) but towards the end of the paper, it was mentioned that using games to train laparoscopic skills may not be efficient (discussion, line 417-426). In addition, it is not fair to judge a person’s potential of laparoscopic skill through the use of video games as both of these skills require different skill sets. Although the study did manage to find a possible correlation between laparoscopic surgery and computer games capabilities, it will be more purposeful if there is a coherent purpose. One suggestion could be finding out that through the use of video games, one is able to improve the 

critical thinking of the individual due to the short amount of time given to react to events in the game. This finding could result to a possible change to how laparoscopic simulations are conducted. For example, having time limits and creating a stressful environment as used for the custom games (Definition of the requirements for custom video games, line 112-113). 

AUTHORS’ Response to Reviewer 1, Comment 2:

We made this more clear

- - - - - - - - - - - - - - - - - - - - - - - - - - - - - - - - - - - - - - - - - - - - - - - - - - - - - - - - - - -

Reviewer 1, Comment 3:

Adding in an additional pool of subjects 

One interesting suggestion is to have the professional Esports players participate in the laparoscopic task. The findings might give more insights for this study as the professional Esports players are definitely considered as experienced players for the comparison of the study. And this could further substantiate the intended aim of the study which is to find the connection between the level of both the potential laparoscopic surgery abilities and gaming skill. 

AUTHORS’ Response to Reviewer 1, Comment 3:

We agree with Reviewer 1 that a future study could investigate the laparoscopic performance of professional gamers, such as E-Sports players. However, this was beyond the scope of this investigation. Because the E-Sports players were recruited at a special event, it is not possible for us to extend the present study with a group of E-sports players performing the laparoscopic tasks.

- - - - - - - - - - - - - - - - - - - - - - - - - - - - - - - - - - - - - - - - - - - - - - - - - - - - - - - - - - -

Reviewer 2, Comment 1:

Throughout the manuscript, references are made to the terms “skill”, “experience” and “performance”. If they refer to the same characteristic, perhaps the authors would like to stick to a single term. Otherwise, defining each term would help with the clarity of the text. In a similar vein, the authors reference the inaccuracy of equating “gaming experience” with “gaming skill” in the introduction. However, the title of the manuscript references “gaming skill” while the text in the abstract makes references to “gaming experience”. Likewise in the discussion, these terms seem to be used interchangeably.

AUTHORS’ Response to Reviewer 2, Comment 1:

We thank Reviewer 2 for this suggestion. We have revised the terminology and now only refer to gaming skills.

- - - - - - - - - - - - - - - - - - - - - - - - - - - - - - - - - - - - - - - - - - - - - - - - - - - - - - - - - - -

Reviewer 2, Comment 2:

The choices in the type of games seem quite arbitrary. For example, why was a 2D side-scroller used and is this the best type of game to measure eye-hand coordination?

A number of known factors contributing to difficulty of laparoscopic surgery are inverted tool motion due to fulcrum effects of the tool ports, loss of 3D vision and disconnected viewpoints from the laparoscopic camera. Why wasn’t established games with similar difficulties utilized instead? Alternatively, using input devices emulating the difficulties present in laparoscopic surgery would have been more appropriate. For example, perhaps using a controller with dual thumbsticks instead of mouse/arrow keys could be more representative of their ability to adapt to hand-eye coordination skills.

AUTHORS’ Response to Reviewer 2, Comment 2:

- - - - - - - - - - - - - - - - - - - - - - - - - - - - - - - - - - - - - - - - - - - - - - - - - - - - - - - - - - -

Reviewer 2, Comment 3:

More details of the background of the professional gamers should be included and controlled for. ESL spans a large range of game genres. A professional real-time strategy or a first person shooter gamer would have different skillsets that they each excels in and measure differently when tested with the authors’ games.

AUTHORS’ Response to Reviewer 2, Comment 3:

- - - - - - - - - - - - - - - - - - - - - - - - - - - - - - - - - - - - - - - - - - - - - - - - - - - - - - - - - - -

Reviewer 2, Comment 4:

Were the students controlled for prior experience of laparoscopic surgery? For example, knowledge of laparoscopic surgery, its difficulties and possible experience through videos of procedures or even personal attempts with box trainers.

AUTHORS’ Response to Reviewer 2, Comment 4:

- - - - - - - - - - - - - - - - - - - - - - - - - - - - - - - - - - - - - - - - - - - - - - - - - - - - - - - - - - -

Reviewer 2, Comment 5:

Please cite references in order of appearance in text - [39] in line 382 is referenced out of sequence. Reference [28] in line 429 may be inappropriate. Please ensure if the results of the referenced work supports the statement adequately. In-text citation of references [29-32] are missing.

AUTHORS’ Response to Reviewer 2, Comment 5:

---

## [Decision Letter · Decision Letter 1]

26 Feb 2020

PONE-D-19-23296R1

Are Gamers better laparoscopic surgeons? Impact of Gaming Skills on Laparoscopic Performance in “Generation Y” students

PLOS ONE

Dear Dr. Chon,

Thank you for submitting your manuscript to PLOS ONE. After careful consideration, we feel that it has merit but does not fully meet PLOS ONE’s publication criteria as it currently stands. Therefore, we invite you to submit a revised version of the manuscript that addresses the points raised during the review process.

We would appreciate receiving your revised manuscript by Apr 11 2020 11:59PM. To enhance the reproducibility of your results, we recommend that if applicable you deposit your laboratory protocols in protocols.io, where a protocol can be assigned its own identifier (DOI) such that it can be cited independently in the future. For instructions see: http://journals.plos.org/plosone/s/submission-guidelines#loc-laboratory-protocols

We look forward to receiving your revised manuscript.

Kind regards,

Chee Kong Chui, PhD

Academic Editor

PLOS ONE

Additional Editor Comments (if provided):

The authors has responded to the question raised by the reviewers. Nevertheless, the responses should be added into the main text.

Reviewers' comments:

Reviewer's Responses to Questions

**Comments to the Author**

1. If the authors have adequately addressed your comments raised in a previous round of review and you feel that this manuscript is now acceptable for publication, you may indicate that here to bypass the “Comments to the Author” section, enter your conflict of interest statement in the “Confidential to Editor” section, and submit your "Accept" recommendation.

Reviewer #1: All comments have been addressed

2. Is the manuscript technically sound, and do the data support the conclusions?

Reviewer #1: Yes

3. Has the statistical analysis been performed appropriately and rigorously? 

Reviewer #1: Yes

4. Have the authors made all data underlying the findings in their manuscript fully available?

Reviewer #1: Yes

5. Is the manuscript presented in an intelligible fashion and written in standard English?

Reviewer #1: Yes

6. Review Comments to the Author

Reviewer #1: Thank you for addressing to the comments. The paper is better understood with the help of the answers and thus, would like to see the answers added into the paper to make everyone understand it. For example, adding on the premise of obtaining the professional gamers and some examples of the games they play. This could further substantiate the validity of the custom games.

7. PLOS authors have the option to publish the peer review history of their article (what does this mean?). If published, this will include your full peer review and any attached files.

Reviewer #1: No

---

## [Author Response · Author response to Decision Letter 1]

9 Apr 2020

Dear Prof. Kong Chui,

Thank you very much for giving us the opportunity to resubmit our revised version of the manuscript. As suggested by Reviewer 1, we have incorporated our answers to the questions raised by the reviewers in the last round of reviews into the manuscript.

- - - - - - - - - - - - - - - - - - - - - - - - - - - - - - - - - - - - - - - - - - - - - - - - - - - - - - - - - - -

Reviewer 1, Comment 1:

Thank you for addressing to the comments. The paper is better understood with the help of the answers and thus, would like to see the answers added into the paper to make everyone understand it. For example, adding on the premise of obtaining the professional gamers and some examples of the games they play. This could further substantiate the validity of the custom games.

AUTHORS’ Response to Reviewer 1, Comment 1:

We thank Reviewer 1 for this suggestion. We have added our answers from the last round of reviews to the Discussion of the manuscript to increase the understanding of our study.

- - - - - - - - - - - - - - - - - - - - - - - - - - - - - - - - - - - - - - - - - - - - - - - - - - - - - - - - - - -

---

## [Decision Letter · Decision Letter 2]

28 May 2020

PONE-D-19-23296R2

Are Gamers better laparoscopic surgeons? Impact of Gaming Skills on Laparoscopic Performance in “Generation Y” students

PLOS ONE

Dear Dr. Chon,

Thank you for submitting your manuscript to PLOS ONE. After careful consideration, we feel that it has merit but does not fully meet PLOS ONE’s publication criteria as it currently stands. Therefore, we invite you to submit a revised version of the manuscript that addresses the points raised during the review process.

We look forward to receiving your revised manuscript.

Kind regards,

Chee Kong Chui, PhD

Academic Editor

PLOS ONE

Additional Editor Comments (if provided):

Please perform a Student's t-test in the revised version of the paper according to the comments of the statistical reviewer.

Reviewers' comments:

Reviewer's Responses to Questions

**Comments to the Author**

1. If the authors have adequately addressed your comments raised in a previous round of review and you feel that this manuscript is now acceptable for publication, you may indicate that here to bypass the “Comments to the Author” section, enter your conflict of interest statement in the “Confidential to Editor” section, and submit your "Accept" recommendation.

Reviewer #3: (No Response)

2. Is the manuscript technically sound, and do the data support the conclusions?

Reviewer #3: Yes

3. Has the statistical analysis been performed appropriately and rigorously? 

Reviewer #3: Yes

4. Have the authors made all data underlying the findings in their manuscript fully available?

Reviewer #3: Yes

5. Is the manuscript presented in an intelligible fashion and written in standard English?

Reviewer #3: Yes

6. Review Comments to the Author

Reviewer #3: From a statistical point of view, the analysis is well-done. Nevertheless, a check of the underlying assumptions of all the statistical methods considered (such as Student’s t-tests and mixed ANOVAs) is missing. Such a check needs to be added in the revised version of the paper. Once the check is done, if the underlying assumptions do not hold on the available data, a nonparametric version of the considered methods, making fewer assumptions, may be considered.

7. PLOS authors have the option to publish the peer review history of their article (what does this mean?). If published, this will include your full peer review and any attached files.

Reviewer #3: No

---

## [Author Response · Author response to Decision Letter 2]

15 Jun 2020

Dear Prof. Kong Chui,

Thank you very much for giving us the opportunity to resubmit our revised version of the manuscript. The underlying assumptions were tested for all statistical tests. If the assumptions were not met, nonparametric tests or corrections were used. The corrections are now also reported in the manuscript. The results of all statistical tests (significance vs. non-significance) remained the same and the conclusions of the paper are unaffected. Please find attached the revised manuscript with and without track changes.

---

## [Decision Letter · Decision Letter 3]

13 Jul 2020

Are Gamers better laparoscopic surgeons? Impact of Gaming Skills on Laparoscopic Performance in “Generation Y” students

PONE-D-19-23296R3

Dear Dr. Chon,

We’re pleased to inform you that your manuscript has been judged scientifically suitable for publication and will be formally accepted for publication once it meets all outstanding technical requirements.

Kind regards,

Chee Kong Chui, PhD

Academic Editor

PLOS ONE

Additional Editor Comments (optional):

I appreciate your efforts in addressing all our concerns. I believe this will be a paper that will be of interests to many readers of this journal.

Reviewers' comments:

Reviewer's Responses to Questions

**Comments to the Author**

1. If the authors have adequately addressed your comments raised in a previous round of review and you feel that this manuscript is now acceptable for publication, you may indicate that here to bypass the “Comments to the Author” section, enter your conflict of interest statement in the “Confidential to Editor” section, and submit your "Accept" recommendation.

Reviewer #3: All comments have been addressed

2. Is the manuscript technically sound, and do the data support the conclusions?

Reviewer #3: Yes

3. Has the statistical analysis been performed appropriately and rigorously? 

Reviewer #3: Yes

4. Have the authors made all data underlying the findings in their manuscript fully available?

Reviewer #3: Yes

5. Is the manuscript presented in an intelligible fashion and written in standard English?

Reviewer #3: Yes

6. Review Comments to the Author

Reviewer #3: (No Response)

7. PLOS authors have the option to publish the peer review history of their article (what does this mean?). If published, this will include your full peer review and any attached files.

Reviewer #3: No

---

## [Editor Report · Acceptance letter]

28 Apr 2020

PONE-D-19-23296R2 

Are Gamers better laparoscopic surgeons? Impact of Gaming Skills on Laparoscopic Performance in “Generation Y” students 

Dear Dr. Chon:

I am pleased to inform you that your manuscript has been deemed suitable for publication in PLOS ONE. Congratulations! Your manuscript is now with our production department. 

With kind regards,

on behalf of

Dr. Chee Kong Chui 

Academic Editor

PLOS ONE